


1 Vertical profiles of sediment methanogenic potential and communities in two plateau

2 freshwater lakes

5        Yuyin Yang[1], Bingxin Li[1], Shuguang Xie[1,*], Yong Liu[2,*]

[1]State Key Joint Laboratory of Environmental Simulation and Pollution Control,
College of Environmental Sciences and Engineering, Peking University, Beijing
100871, China
[2]Key Laboratory of Water and Sediment Sciences (Ministry of Education), College of
Environmental Sciences and Engineering, Peking University, Beijing 100871, China
* Corresponding author. Tel: 86-10-62751923. Fax: 86-10-62751923.
Email: xiesg@pku.edu.cn (S Xie); yongliu@pku.edu.cn (Y Liu)



**Abstract**
Microbial methanogenesis in sediment plays a crucial role in $CH_4$ emission from
freshwater lake ecosystem. However, knowledge on the layer depth-related changes
of methanogens and their activities in freshwater lake sediment is still limited. The
present study was conducted to characterize the methanogenesis potential in different
sediment layer depths and the vertical distribution of microbial communities in two
freshwater lakes at different trophic status on the Yunnan Plateau (China). Incubation
experiments and inhibitor studies were carried out to determine the methanogenesis
potential and pathways. *McrA* and 16S rRNA genes were used to investigate the
abundance and structure of methanogen and archaeal communities, respectively.
Hydrogenotrophic methanogenesis was mainly responsible for methane production in
sediments of both freshwater lakes. The layer depth-related change pattern of the
methanogenesis potential in Dianchi Lake was found to be different from that in Erhai
Lake. *mcrA* and archaeal 16S rRNA genes displayed the similar abundance change
pattern in either lake, and the relative abundance of methanogens decreased with
increasing sediment layer depth. Archaeal communities differed considerably in
Dianchi Lake and Erhai Lake, but methanogen communities showed a slight
difference between in these two lakes. However, methanogen communities illustrated
a remarkable layer depth-related change. Order *Methanomicrobiales* was the
dominant methanogen group in all sediments, while *Methanobacteriales* showed high
proportion only in upper layer sediments.





**Keywords:** Methanogenesis pathways; Freshwater lake sediment; Vertical profile;
*Archaea*; *mcrA*






















1. **Introduction**
Methane ($CH_4$) is an effective greenhouse gas in atmosphere, and lacustrine
ecosystems may be responsible for 6–16% of natural methane emission (Bastviken et
al., 2004). In anoxic sediment of freshwater lake, a large amount of methane can be
produced through microbial methanogenesis (Bastviken et al., 2008; Gruca-Rokosz
and Tomaszek, 2015). Biogenic methane is produced by the activity of methanogens,
a strictly anaerobic microbial group belonging to archaeal phylum *Euryarchaeota*.
Methanogens from various archaeal orders have been reported (Garcia, 1990; Paul et
al., 2012; Sakai et al., 2008), and their substrate is generally the end products of
organic matter degradation by fermentative bacteria and archaea (Borrel et al., 2011).
Methanogens produce methane through either hydrogenotrophic (using $H_2/CO_2$) or
acetoclastic pathway (using acetate, i.e. the methyl group) (Conrad et al., 2010). To
determine the contribution of both methanogenic pathways, approaches including
isotope labeling, $\delta^{13}C$ analysis and inhibitor study have been applied (Conrad, 2005).
Despite the theoretical ratio of 2:1 (acetoclastic pathway: hydrogenotrophic pathway)
(Conrad, 1999), most methane in freshwater lake is produced through
hydrogenotrophic pathway (Borrel et al., 2011). However, the relative significance of
hydrogenotrophic pathway remains unclear, because it can vary considerably with
lake (Conrad, 1999). To identify the methanogens involved in methanogenesis, both
archaeal 16S rRNA gene and functional *mcrA* gene have been used (Conrad et al.,
2007; Luton et al., 2002; Orphan et al., 2008). The dominance of *Methanomicrobiales*
and *Methanosarcinale* have been reported in a variety of freshwater lakes (Biderre-

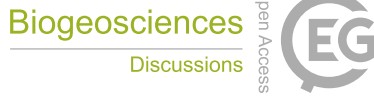
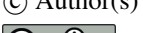

Petit et al., 2011; Conrad et al., 2007; Youngblut et al., 2014). In freshwater lakes,
both methanogenesis pathway and methanogenic community structure can change
with sediment layer depth (Chan et al., 2005; Liu et al., 2016; Lofton et al., 2015).

Many previous studies have investigated methanogenesis in humic lakes (Youngblut
et al., 2014), oligotrophic lakes (Lofton et al., 2015), and meromictic lakes (Biderre-
Petit et al., 2011; Gies et al., 2014), while shallow meso- and eutrophic lakes have
attracted poor attention. Substrate plays an important role in methanogenesis, and has
a considerable effect on pathway preference (Liu et al., 2016; Nozhevnikova et al.,
2007). Methanogenesis in meso- and eutrophic lakes that have abundant substrate
might be different from that in other previously studied lakes. Therefore, the
following questions attracted our attention: in sediments of mesotrophic and eutrophic
lakes, (1) How do the methanogenesis activity and contribution of different pathways
change with the increasing sediment layer depth? (2) How do methanogen community
structure and the dominant methanogens change along the sediment layer depth? (3)
Are these changing patterns similar in lakes at different trophic status?

2. **Materials and methods**
2.1. *Sampling sites and samples*
Dianchi Lake is a highly eutrophic lake with an area of 309 $km^2$ and the average water
depth of 4.4 m, while mesotrophic Erhai Lake has an area of 250 $km^2$ and the average
water depth of 10 m (Wang et al., 2015). In this study, five replicate sediment cores



(0–20 cm) were collected at the profundal area of both Dianchi Lake and Erhai Lake
with a columnar sediment sampler in December 2015. The water depths at the
sampling sites were 6.4 and 11 m in Dianchi Lake and Erhai Lake, respectively. The
in-situ sediment temperature were 16.4ºC in Dianchi Lake and 14.8 ºC in Erhai Lake,
respectively. Meanwhile, a total of 500 mL bottom water at each sampling site was
collected for the subsequent incubation experiments.

The five replicate sediment cores were sliced into the layers: 0–5, 5–8, 8–11, 11–14,
14–17, and 17–20 cm. Replicate sediment samples from the same layer depth in a
given lake were mixed and then subsampled for physicochemical and molecular
analyses and incubation experiments. Sediment samples for physicochemical and
molecular analyses were stored at -20ºC, while those for incubation experiments were
kept in gas-tight bottles under anoxic condition at 4ºC. The sediment samples were
transported to laboratory within one week, which would have no considerable effect
on methanogenic activity (Nüsslein et al., 2001). The levels of sediment total organic
carbon (TOC), total nitrogen (TN), nitrate nitrogen ($NO_3^-$-N), ammonium nitrogen
($NH_4^+$-N), and total phosphorus (TP) were shown in Figure S1.

2.2. *Methane production potential measurement*
The incubation experiments for methane production potential (MPP) measurement
were performed with reference to the standard procedure described in the literature
(Conrad et al., 2010). Uppermost sediment (0–5 cm) was centrifuged at 4000 rpm for



5 min to obtain the similar water content with the samples from other layers. For each
sediment layer, a total six sediment samples (1.5 g) and 8 mL bottom water were
transferred into a 50-mL sterile serum bottle, flushed with $N_2$, and then closed with a
butyl rubber stopper. After incubation at $16^oC$ overnight, the bottles were flushed with
$N_2$ again, and half of them were added with $CH_3F$ (1 mL) to inhibit the aceticlastic
methanogenesis. Incubation was carried out at $16^oC$ at 100 rpm for 28 days. At the
end of incubation, gas samples were taken from the headspace using a gas-tight
syringe, and then were analyzed using GC126 gas chromatography (INESA
instrument, Shanghai) with a flame ionization detector.

2.3 *Molecular analysis*
DNA was extracted using the Powersoil DNA extraction kit (Mobio Laboratories,
USA). The quality of DNA were checked using 1.0% agarose gel electrophoresis. For
quantitative PCR (qPCR), the primer sets mcrF (5'-
GGTGGTGTMGGATTCACACARTAYGCWACAGC-3') /mcrR (5'-
TCATTGCRTAGTTWGGRTAGTT-3') (Luton et al., 2002) and Arch344F (5'-
GYGCAGCAGGCGCGA-3') /Arch915R (5'-GTGCTCCCCCGCCAATTCCT-3')
(Casamayor et al., 2002; Conrad et al., 2014) were used for the quantification of
archaeal 16S rRNA and *mcrA* genes, respectively. The qPCR assay was carried out
using an ABI 7500 FAST (Applied Biosystems).The reaction mixture included
2×SYBR Green PCR master mix (12.5 μL), 10 μM of each primer (1 μL), and
template DNA (2 ng). The PCR conditions were as described in the literatures





(Casamayor et al., 2002; Luton et al., 2002). Standard curves ranging from $10^3$ to $10^7$
gene copies/mL were generated using serial dilutions of linearized plasmids (pGEM-
T, Promega) containing cloned target gene amplified from environmental DNA. The
coefficient ($r^2$) for archaeal 16S rRNA gene and *mcrA* gene were 0.9995 and 0.9998,
respectively. Significance was tested using one-way analysis of variance (ANOVA),
followed by S-N-K's post-hoc analysis (when the variances were homogenous) or
Dunnett's post-hoc analysis (when the variances were not homogenous).

For Illumina MiSeq sequencing, archaeal 16S rRNA gene was amplified using primer
set Arch519f (5'-CAGCCGCCGCGGTAA-3')/Arch915R (5'-
GTGCTCCCCCGCCAATTCCT-3') (He et al. 2016; Herfort et al., 2009; Long et al.
2016). The PCR products from triplicate samples were mixed in equal amounts and
were subjected to Illumina MiSeq sequencing. Raw reads were merged and the
quality filtering was carried out using Trimmomatic (Bolger et al., 2014) and FLASH
(Magoc and Salzberg, 2011). The obtained raw reads were deposited in the NCBI
SRA (short-read archive) under accession SRP076837. After subsampling to the
lowest number of sequences, sequences from each sediment sample were grouped into
OTUs (operational taxonomic units) using Usearch (version 7.1,
http://drive5.com/uparse/) at 97% similarity. Diversity analysis was carried out using
the MOTHUR program (version v.1.30.1) (Schloss et al., 2011). The taxonomic
identities of representative sequences for each OTU were assigned using the Silva 16S
rRNA database (Quast et al., 2013). For *mcrA* gene clone library analysis, primer set



mcrF/mcrR was used, and the PCR conditions were as previously described (Luton et
al., 2002). The obtained *mcrA* sequences were deposited in the GenBank database
under accession numbers KU997795-KU997842, KX196972-KX197020 and
KX093502-KX093920. The chimera-free *mcrA* gene sequences were grouped into
OTUs at the similarity level of 89% (Webster et al., 2014) using the MOTHUR
program. OTU-based diversity was also calculated using the MOTHUR program
(Schloss et al., 2009). Phylogenetic analysis of the *mcrA* gene sequences was carried
out with the MEGA 6.0 software (Tamura et al., 2013). Weighted Unifrac distance
between samples was calculated using *R* library GUniFrac, and PCoA (Principal
Coordinate Analysis) and environment clusters analysis were conducted based on
Weighted Unifrac distance using R (version i386, 3.3.0).

3. **Results**
3.1. *Methane production potential*
In this study, MPP varied remarkably with both lake and sediment layer depth (Figure
1). In contrast, MPP differed slightly in the uppermost layer sediments (0–5 cm) of the
studied two lakes. The change pattern of MPP differed in the two lakes. In Dianchi
Lake, the MPP generally increased with increasing layer depth (varying from 2.46 to
11.26 nmol/gDW/day in 5–20 cm layer), while in Erhai Lake, the MPP decreased
from 2.13 to 0.40 nmol/gDW/day. The sediment samples from Dianchi Lake showed a
significantly higher MPP than those from Erhai Lake ($P<0.05$).





The hydrogenotrophic methanogenesis potential (HMP) was measured under the
condition of 2% $CH_3F$. HMP showed a similar change trend with MPP in either of the
studied two freshwater lakes. With increasing layer depth, MPP generally increased in
Dianchi Lake but tended to decrease in Erhai Lake. Dianchi Lake had the HMP of
1.68–6.53 nmol/gDW/day, but Erhai Lake had a much lower HMP (0.34–1.36
nmol/gDW/day). Besides, methane production through aceticlastic pathway varied
greatly in different layers (0.07–3.00 nmol/gDW/day in Dianchi Lake, 0.04–0.49
nmol/gDW/day in Erhai Lake), and showed a notable difference between in these two
freshwater lakes.

3.2. *Community abundance of Archaea and methanogen*
The depth-related change pattern of either archaeal 16S rRNA or *mcrA* gene differed
in Dianchi Lake and Erhai Lake (Figures 2a and 2b). The density of archaeal 16S
rRNA gene fluctuated between $2.9 \pm 0.1 \times 10^7$ and $8.4 \pm 0.8 \times 10^8$ copies/g dry weight in
Dianchi Lake. However, in Erhai Lake, the archaeal community abundance ranged
from $2.7 \pm 0.1 \times 10^7$ to $3.4 \pm 0.1 \times 10^8$ copies/g dry weight, and showed an increase
followed by a significant decrease ($P<0.05$), with the peak value in the 5–8-cm layer.

As for *mcrA* gene, its highest density was observed at the uppermost layer (0–5 cm) in
either Dianchi Lake ($1.06 \pm 0.06 \times 10^6$ copies/g dry weight) or Erhai Lake ($5.6 \pm$
$0.2 \times 10^5$ copies/g dry weight), while the lowest one occurred in Dianchi Lake sample
D6 ($2.5 \pm 0.3 \times 10^4$ copies/g dry weight) or Erhai Lake sample E5 ($3.7 \pm 0.1 \times 10^4$ copies/g





dry weight). The gene abundance generally decreased with increasing layer depth. In
addition, either *Archaea* or methanogens showed greater abundance in Dianchi Lake
than in Erhai Lake.

3.3. *Diversity of archaeal and methanogen communities*
In this study, after normalization, a total of 16,028 archaeal sequences were retrieved
from each sediment sample. The number of OTUs in each library ranged between 547
and 1194, and library coverage was 98.07–99.64% (Table 1). Archaeal community
diversity of most sediment samples varied slightly (Shannon index=4.51–5.13),
whereas samples D2 and D3 had much lower Shannon diversity (3.98 or 3.88). The
samples from Erhai Lake showed higher archaeal diversity (Shannon index=4.81–
5.13) than those from Dianchi Lake (Shannon index=3.88–4.75).

A total of 516 *mcrA* gene sequences were retrieved from *mcrA* clone library, and
could be assigned into 30 OTUs (at 89% similarity). For each sample, the number of
sequences ranged from 35 to 49, while the number of OTUs varied between 6 and 11.
The coverage for each clone library was no less than 85.7%, indicating that the *mcrA*
OTUs of each sample had been well captured. The Shannon indices were 1.49–1.72
for Dianchi Lake and 1.25–1.84 for Erhai Lake, respectively. In Dianchi Lake, sample
D1 had the highest *mcrA* gene diversity, but sample D6 had the lowest diversity. In
Erhai Lake, sample E2 had the highest *mcrA* gene diversity followed by sample E1,
but E5 showed the lowest diversity.




### 3.4. *16S rRNA-based community composition*

*Bathyarchaota* (formally known as *Miscellaneous Crenarchaeotic_Group*, *MCG*) and

*Euryarchaeota* dominated the archaeal communities in both Dianchi Lake and Erhai

Lake (Figure 3). In the uppermost sediment layer in either lake, *Bathyarchaota*

accounted for nearly 20% of the total archaeal community. However, in Dianchi Lake,

with increasing layer depth, the proportion of *Bathyarchaota* showed a remarkable

decrease followed by a rise. *Bathyarchaota* organisms showed high proportion (44–

55%) in Dianchi Lake samples D4–D6 and Erhai Lake samples E2–E6. In contrast,

*Euryarchaeota* illustrated an opposite change trend. In Dianchi Lake, the proportion

of *Euryarchaeota* organisms was higher in samples D2 and D3 (63% or 67%) than in

sample D1 (47%), but displayed a notable decrease in deeper layers (26–27%). In

Erhai Lake, the *Euryarchaeota* proportion in sample E1 was 32.5% but became lower

in samples E2–E6 (9–23%). In addition, *Woesearchaeota* showed a greater abundance

in Dianchi Lake (8–13%) than in Erhai Lake (4–9%), while *Crenarchaeota* was

mainly distributed in Erhai Lake (3–29%).

259

Based on 16S rRNA gene analysis, methanogens comprised 3–17% of the total

archaeal community (Figure S2). The relative abundance of methanogens tended to

decrease with increasing sediment layer depth. Order *Methanobacteriales* dominated

the methanogen community in the uppermost layer samples (samples D1 or E1), and

also had a considerable proportion in samples D2 and D3. In contrast,



*Methanomicrobiales* was the dominant methanogen group in the lower layer
sediments (Dianchi Lake samples D2–D6 and Erhai Lake samples E2–E6).
*Methanocellales* and *Methanosarcinales* could also be detected in all sediment
samples, but their proportions were less than 2%. At genus level, *Methanobacterium*
had the greatest proportion, followed by *Methanosaeta* and *Methanoregula*, and the
proportion of each archaeal genus decreased with increasing layer depth in either lake
(data not shown).

3.5. *mcrA-based community composition*
In the present study, the 516 *mcrA* sequences from all sediment samples fell into 30
OTUs, 12 of which had only one sequence. The representative sequences from the
OTUs with at least two sequence members were further used to construct the
phylogenic tree with their close *mcrA* sequences reported in the NCBI database
(Figure 4). All of the sequences were grouped into five clusters (clusters 1−5). Cluster
1 contained most of the obtained *mcrA* sequences (440), and could be affiliated with
the *mcrA* sequences from *Methanomicrobiales*. Cluster 2 consisted of 29 *mcrA*
sequences related to those from *Thermoplasmatales* and *Methanoplasmatales*. Cluster
3 (with 13 sequence members) and cluster 5 (with 12 sequence members) were related
to *Methanosarcinale* and *Methanobacteriales*, respectively. Cluster 4 was the smallest
group and only contained 10 sequences. The sequences in this cluster were not related
to the *mcrA* sequences from known methanogens. In addition, the sequences affiliated
to cluster 1 could be further divided into 3 clades. OTU7 was grouped together with



the sequences from two *Methanomicrobiaceae* strains. OTU8 showed a close relation
to a *Methanolinea*-like *mcrA* sequence, and OTU1, OTU2 and OTU6 were closely
related to *Methanoregula*-like *mcrA* sequence.

*Methanoregula*-like *mcrA* sequences (OTU1 and OTU2) were detected in all sediment
layer depths, but the proportion of OTU1 decreased with increasing layer depth
(Figure S3). *Methanomicrobiaceae*-like *mcrA* sequences (OTU7) were not detected in
Dianchi Lake samples D1, D2 and D3, but they dominated in the other samples and
their proportion increased with increasing sediment layer depth. *Methanoplasmatales*
-like *mcrA* sequences (OTU13) mainly existed in Dianchi Lake, but were also
detected in the uppermost sediment layer in Erhai Lake. *Methanolinea*-like *mcrA*
sequences (OTU8) were present in all sediment layers in both lakes, but their change
pattern was not evident. In addition, other genotypes of *mcrA* gene sequences
comprised less than 22% of the total sequences in each sample.

3.6. *Comparison of archaeal and methanogen communities*
The difference among archaeal assemblages was discriminated using Weighted-
Unifrac distance-based cluster analysis (Figure 5a). In either Dianchi Lake or Erhai
Lake, the sample from the uppermost layer (sample D1 or E1) was separated from the
samples in other sediment layers. In Dianchi Lake, samples D2 and D3 were also
grouped into one clade, and the other three samples fell into another one. However, in
Erhai Lake, samples E2, E3, E4, E5 and E6 were grouped together, and the samples





from neighboring layers tended to have a relatively similar archaeal community
structure.

For methanogen communities, the studied 12 sediment samples fell into two groups
(Figure 5b). Samples D1, D2, D3 and E1 were clustered together. For other samples,
samples at similar layer depth tended to have relatively similar methanogen
community structure. Moreover, the sediment samples from two lakes were not
clearly separated.

4. **Discussion**
4.1 *Methane production potential in freshwater lake sediment*
The methanogenesis potential varied in a wide range (Dan et al., 2004; Duc et al.,
2010; Lofton et al., 2015), from less than 1 nmol $CH_4$/gDW day to more than one
thousand nmol $CH_4$/gDW day, and the methanogenesis potential obtained in the
current study fell in this range. Numerous previous studies showed that MPP
decreased with increasing sediment layer depth (Chan et al., 2005; Liu et al., 2016;
Lofton et al., 2015), or displayed a slight shift followed by a sharp decrease (Lofton et
al., 2015). The change pattern of MPP in Erhai Lake was similar to that reported in
these literatures. However, to the authors' knowledge, Dianchi Lake was the first lake
that illustrated the increase of MPP with increasing sediment layer depth.

Several previous studies had investigated the inter-lake difference of methanogenesis.



Methane production rate could differed drastically between in two
geomorphologically similar oligotrophic lakes, and the quantity and quality of water
dissolved organic carbon (DOC) might be an influential factor (Lofton et al., 2015).
Other environmental factors, including geological constitute, geographical regions
(Rinta et al., 2015) as well as water type (Conrad et al., 2014), were also found to
have considerable influences on methane production rate in freshwater lake sediment.
Moreover, lake characteristics could influence the methane production rate both
directly and indirectly (Borrel et al., 2011). In this study, the methanogenesis potential
in Dianchi Lake was found to be much higher than that in Erhai Lake. Dianchi Lake
had a larger amount of nutrient than Erhai Lake, and a high level of organic matter
could exist in lower sediment layers (Figure S1). MPP was found to be related to the
availability of organic matter (Liu et al., 2016; Lofton et al., 2015; Nozhevnikova et
al., 2007), so the abundant substrate in Dianchi Lake could favor higher MPP. In
addition, although methanogen communities in Dianchi Lake and Erhai Lake had
similar structure, either archaeal or bacterial community differed greatly in these two
lakes (Dai et al., 2016; Yang et al., 2016). Bacterial and non-methanogen archaeal
community played important roles in decomposing organic matter, and thus
influenced the availability of substrate.

By comparing the uninhibited and the inhibited methanogenesis potential, it could be
inferred that hydrogenotrophic pathway played a major role in Dianchi Lake (75.8 %
of total methane production potential). The result was consistent with the previous



studies (Conrad et al., 2010; Liu et al., 2013; Liu et al., 2016). However, in Erhai
Lake, the methanogenesis potential through aceticlastic pathway was comparable to
that through hydrogenotrophic pathway. Moreover, acetoclastic methangenesis was
found to decrease with increasing lake sediment layer depth (Chan et al., 2005; Liu et
al., 2016), which could be attributed to the decreasing availability of organic matter
(Liu et al., 2016). In this study, the layer depth-related difference in the contribution
of two methanogenic pathways was not clear. This might be partly due to the
fluctuation of acetoclastic methangenesis with sediment layer depth. This also might
suggest the excess substrate for methanogenesis in sediments of eutrophic lake.

4.2. *Abundance of Archaea and methanogen in freshwater lake sediment*
According to the previous studies, the abundance of archaeal 16S rRNA gene
generally decreased with increasing layer depth in stratified lake sediments (Chan et
al., 2005; Zhu et al., 2012). However, in shallow and eutrophic lakes, both archaeal
16S rRNA gene and *mcrA* gene abundance could fluctuate along the sediment depth
gradient (Ye et al., 2009; Zhu et al., 2012). In the current study, the density of archaeal
16S rRNA gene tended to decrease with increasing layer depth in Erhai Lake but
considerably fluctuated in Dianchi Lake, which might be attributed to the difference
of substrate supply and lake water depth.

The abundance of methanogens could be assessed using either archaeal 16S rRNA or
*mcrA* gene. Mthanogen abundance was usually found to rise at first followed by a



decrease with increasing layer depth, and its peak occurred at the surface layer or 5–
10 cm beneath the sediment surface (Borrel et al., 2012; Milferstedt et al., 2010; Zhu
et al., 2012), which consisted with the results found in this study.

In the current study, the abundance of archaeal community was comparable to that
reported in the literatures (Borrel et al., 2012; Zhu et al., 2012). However, for each
sample, the *mcrA*/16S ratio was less than 3%, while the sequences affiliated with
methanogen-like organisms accounted for 3–17% of total 16S rRNA sequences. The
result might suggest either the bias of amplification of *mcrA* gene or the numerical
difference between organisms and functional gene.

4.3. *Diversity of archaeal and methanogen communities in freshwater lake sediment*
The diversity of archaeal community usually displayed a slight fluctuation along the
sediment layer depth gradient (Koizumi et al., 2004; Lim et al., 2011; Nam et al.,
2008). This was in agreement with the result found in this study. To date, the diversity
of *mcrA* gene was unclear, partly due to the usage of order-specific primers of
methanogens in the previous studies (Borrel et al., 2012; Zhu et al., 2012). Moreover,
information on the diversity of methanogen 16S rRNA gene was lacking. In this
study, relatively higher diversity of *mcrA* gene was observed in the samples from
upper layers (sample D1 in Dianchi Lake and sample E2 in Erhai Lake), but the
change pattern of *mcrA* gene diversity was not clear. In addition, both community
diversity and evenness of methanogens could vary with lake (Milferstedt et al., 2010;



Youngblut et al., 2014). In this current study, the sediment samples from Erhai Lake
had slightly lower *mcrA* gene diversity than those from Dianchi Lake.

4.4 *Composition of methanogen community in freshwater lake sediment based on 16S*
*rRNA gene*
Hydrogenotrophic *Methanomicrobiales* was detected in a variety of environments,
and was the most frequently observed archaeal order in freshwater lake sediment
(Biderre-Petit et al., 2011; Youngblut et al., 2014). In the present study,
*Methanomicrobiales* had high proportion in sediment from each layer depth, which
agreed with these two previous studies. Moreover, *Methanobacteriales* was usually
found in the ecosystems with high levels of nutrition and substrate, such as
hypereutrophic Priest Pot (Earl et al., 2003) and eutrophic Taihu Lake (Ye et al.,
2009). In this study, *Methanobacteriales* was mainly distributed in Dianchi Lake
samples D1, D2 and D3 and Erhai Lake sample E1. In addition, methanogens from
order *Methanosarcinales* mainly participated in reducing acetate and methyl
compounds (Borrel et al., 2011) that were relatively abundant and labile in lake
ecosystems. However, *Methanosarcinales* showed lower proportion in Dianchi Lake
and Erhai Lake than in other previous reported freshwater lakes (Biderre-Petit et al.,
2011; Borrel et al., 2012; Zhu et al., 2012).

Ye et al. (2009) documented the layer depth-related change of archaeal community
structure in Taihu Lake, while its change pattern was not clear. In this study, the result





of UniFrac-based cluster analysis indicated that archaeal community structure differed
remarkably in Dianchi Lake and Erhai Lake. In either Dianchi Lake or Erhai Lake,
layer depth was found to be a key determinant to archaeal community structure.
However, the abrupt shift in archaeal community structure occurred at different layer
depths in these two lakes. In Dianchi Lake, the structures of archaeal communities at
layer depth 0–11 cm (including samples D1, D2 and D3) were much different from
those at layer depth 11–20 cm (including samples D4, D5 and D6). However, in Erhai
Lake, a remarkable difference lay between the sample from uppermost layer (D1) and
those from other five layers.

4.5. *Composition of methanogen community in freshwater lake sediment based on*
*mcrA gene*
Based on *mcrA* gene clone library analysis, *Methanomicrobiales* was found to be the
dominant methanogen group in sediments of many freshwater lakes (Biderre-Petit et
al., 2011; Youngblut et al., 2014), which consisted with the result found in this current
study. *Methanoregula* and *Methanolinea*, affiliated within *Methanomicrobiales*, were
among the most frequently detected archaeal genera in freshwater lake (Borrel et al.,
2011). So far, the change pattern of methanogen community at genus level along the
layer depth gradient has not been addressed. In the current study, the proportion of
*Methanoregula*-like *mcrA* sequences tended to decrease with increasing sediment
layer depth, while the proportion of *methanolinea*-like *mcrA* sequences did not show a
clear change pattern. *Methanoplasmatales* -like methanogens were related to





*Thermoplasmatales* archaeons. It was usually present in termite guts and high-salinity
environments (e.g., marine sediment), and was regarded as the seventh order of
methanogens (Paul et al., 2012). Only several previous studies reported their existence
in freshewater lake sediment (Conrad et al., 2014; Liu et al., 2013; Webster et al.,
2014). In this study, *Methanoplasmatales* -like methanogens were detected in both
Dianchi Lake and Erhai Lake, but the distribution pattern along sediment layer depth
in these two lakes was different. *Methanosarcinales* and *Methanobacteriale*s were
also detected in different sediment layers in both lakes. However, their change
patterns were not clear because of their low relative abundance. In addition, compared
with other freshwater lake sediments (Borrel et al., 2012; Zhu et al., 2012), sediments
in Dianchi Lake and Erhai Lake showed much higher *Methanomicrobiales* proportion
but lower *Methanosarcinales* proportion.

The phylogeny of *mcrA* gene was congruent with that of 16S rRNA gene (Luton et al.,
2002; Springer et al., 1995), but in this study, the results based on *mcrA* gene clone
library were not always consistent with that based on16S rRNA Illumina MiSeq
sequencing. For an example, *Methanosarcinales* organisms accounted for 7–20% of
the total methanogens according to 16S rRNA sequencing, but showed a very low
proportion in *mcrA* clone library (less than 3% in 11 out of 12 samples).
*Methanobacteriales* was abundant in uppermost layer sediments based on 16S rRNA
sequencing, but was a minor group in *mcrA* clone library.



Methanogen communities in surface sediment were usually found to be lake-
dependent (Milferstedt et al., 2010; Youngblut et al., 2014), whereas in this study,
UniFrac-based cluster analysis indicated that methanogen communities in sediments
of Dianchi Lake and Erhai Lake were not phylogenetically separated. Sediment layer
depth was found to shape methanogen community structure.

**5. Conclusions**

The MPP and abundance of sediment methanogens differed greatly in Dianchi Lake
and Erhai Lake, while these two lakes had the similar methanogen community
structure, with the dominance of *Methanomicrobiales* and *Methanobacteriales*.
Hydrogenotrophic methanogenesis was the major methane production pathway in
sediments of both lakes. The layer depth-related changes of methanogenesis potential,
and the abundance and community structure of methanogens were observed in either
Dianchi Lake or Erhai Lake. Sediment methanogen community and activity might be
influenced by lake tropic status.

**Acknowledgments**

This work was financially supported by National Natural Science Foundation of
China (No. 41571444) and National Basic Research Program of China
(2015CB458900).




**Competing interests**
The authors declare that they have no conflict of interest.

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






**Table 1** Diversity and library coverage of archaeal 16S rRNA and *mcrA* genes. Samples
D1–D6 and E1–E6 were retrieved from Dianchi Lake and Erhai Lake, respectively.
Digits "1" – "6" were referred to sediment depth 0–5, 5–8, 8–11, 11–14, 14–17 and 17–
20 cm, respectively.

| Sample | Coverage | | Number of OTUs | | Shannon index | |
|---|---|---|---|---|---|---|
| | Arch 16S | *mcrA* | Arch 16S | *mcrA* | Arch 16S | *mcrA* |
| D1 | 98.35% | 97.92% | 1120 | 8 | 4.51 | 1.72 |
| D2 | 98.70% | 91.30% | 897 | 8 | 3.98 | 1.50 |
| D3 | 98.45% | 94.87% | 995 | 6 | 3.88 | 1.54 |
| D4 | 98.38% | 91.43% | 1060 | 8 | 4.73 | 1.59 |
| D5 | 98.14% | 85.71% | 1194 | 11 | 4.75 | 1.65 |
| D6 | 98.53% | 90.00% | 959 | 9 | 4.53 | 1.49 |
| E1 | 98.12% | 93.88% | 1261 | 8 | 5.00 | 1.51 |
| E2 | 98.17% | 90.00% | 1256 | 10 | 5.13 | 1.84 |
| E3 | 98.07% | 95.65% | 1234 | 6 | 4.88 | 1.32 |
| E4 | 98.39% | 93.02% | 1104 | 7 | 4.81 | 1.31 |
| E5 | 98.33% | 93.18% | 1162 | 6 | 4.99 | 1.25 |
| E6 | 99.64% | 95.45% | 547 | 8 | 5.03 | 1.60 |


**Figure captions**

**Figure 1** Uninhibited and 2% $CH_3F$ inhibited methane production potential in different layer depths. *Error bars* represent standard deviation of mean (*n*=3).

**Figure 2** Abundance of archaeal 16S rRNA (a) and *mcrA* (b) genes in different layer depths. *Error bars* represent standard deviation of mean (*n*=3).

**Figure 3** Compositions of archaeal communities at phylum level based on 16S rRNA gene. Samples D1–D6 and E1–E6 were retrieved from Dianchi Lake and Erhai Lake, respectively. Digits "1" – "6" were referred to sediment depth 0–5, 5–8, 8–11, 11–14, 14–17 and 17–20 cm, respectively.

**Figure 4** Phylogenetic tree of representative archaeal *mcrA* sequences and reference sequences from GenBank. The sequences beginning with "D1"–"D6" and "E1"–"E6" were referred to the sequences retrieved from sample D1–D6 and E1–E6, respectively. The bold number in parentheses represents the numbers of the total sequences in the library. Numbers at the nodes indicate the levels of bootstrap support based on neighbor-joining analysis of 1,000 resampled datasets. The values less than 50 are not listed. The bar represents 5% sequence divergence.

**Figure 5** Environment clusters for archaeal 16S rRNA gene (a) and *mcrA* gene (b) assemblages based on Unifrac distance. Samples D1–D6 and E1–E6 were retrieved





from Dianchi Lake and Erhai Lake, respectively. Digits "1" – "6" were referred to

sediment depth 0–5, 5–8, 8–11, 11–14, 14–17 and 17–20 cm, respectively.



**Figure 1**

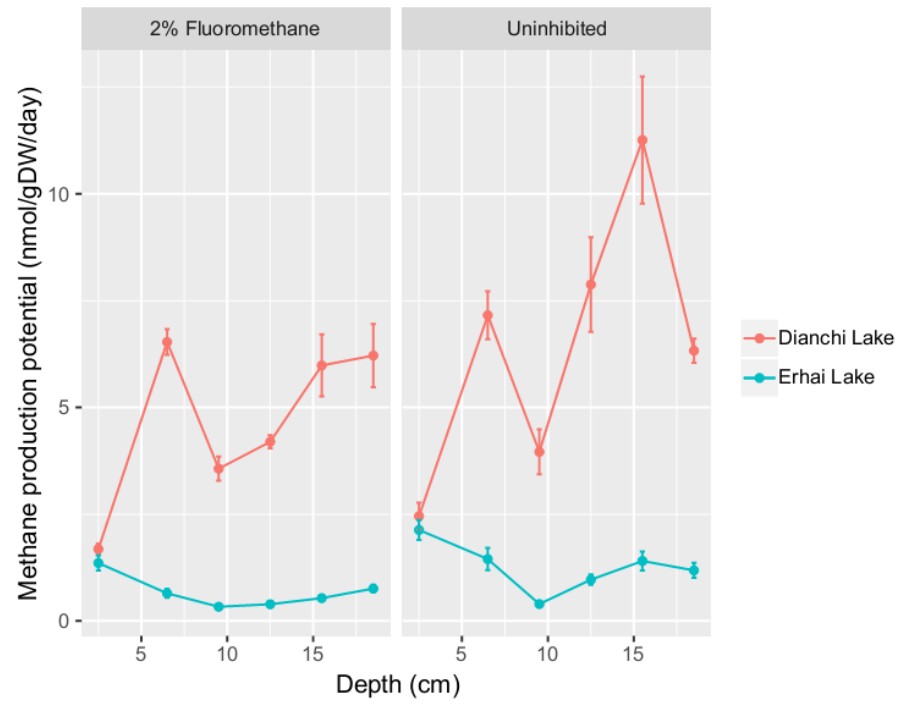





**Figure 2**

**(a)**

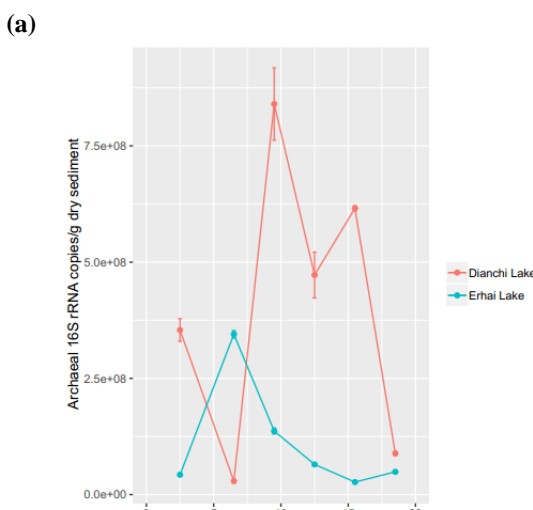

**(b)**

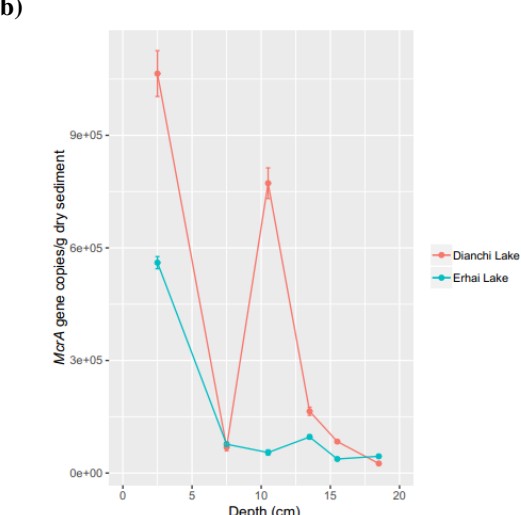




**Figure 3**

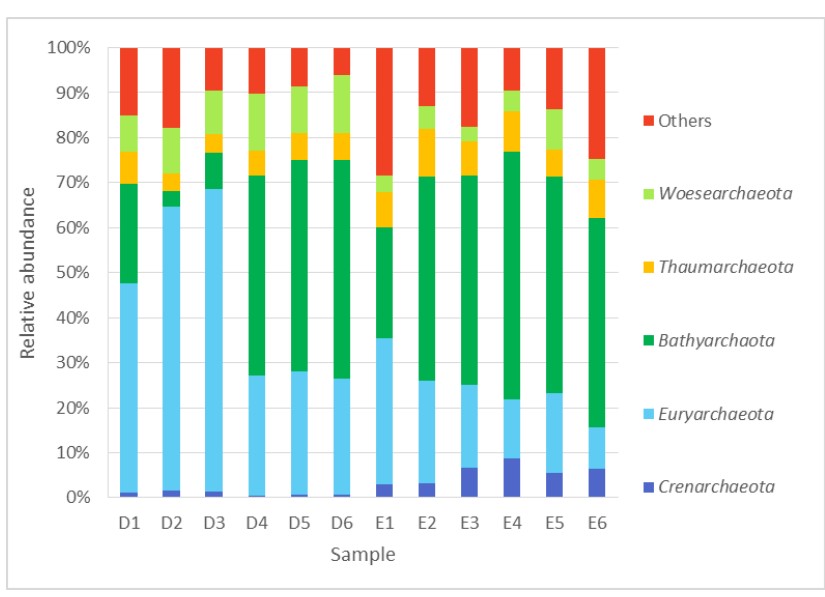





**Figure 4**

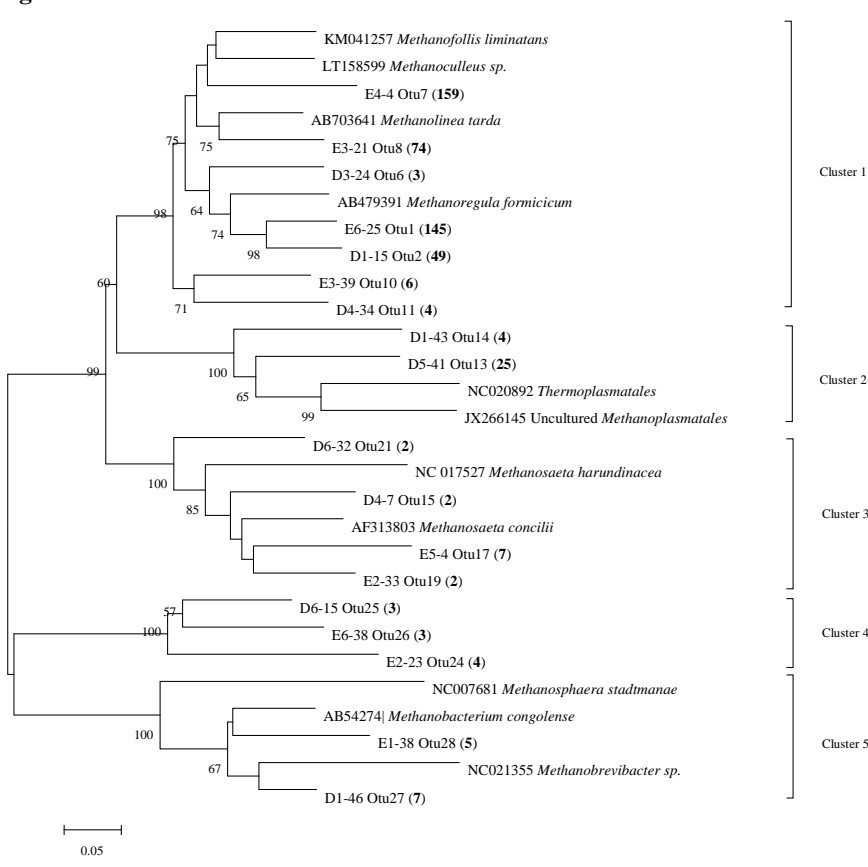



**Figure 5**

**(a)**

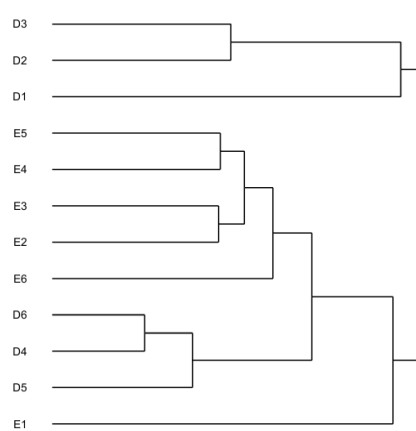

**(b)**