# Peer review of "Vertical profiles of sediment methanogenic potential and communities in two plateau"

_Biogeosciences, 2016_

## Referee Comment (RC1) · Anonymous Referee #1 · 6 Sep 2016

Vertical profile of sediment methanogenic potential and communities in two plateau freshwater lakes

Manuscript Number: bg-2016-321

The manuscript "Vertical profile of sediment methanogenic potential and communities in two plateau freshwater lakes" is comprehensive written. The aim of the study is clearly stated and well supported with data. The authors describe the methanogenic potential (MPP) of sediment incubations, quantify the archaeal and methanogenic community and analyses the community structure using NGS. They can show that the two lakes exhibit different patterns for almost all analyzed parameters and show some changes along a depth profile of 20cm. The MPP measurements would benefit from

<a href="#">Printer-friendly version</a>

<a href="#">Discussion paper</a>

a better time resolution and the additional measurement of the isotopic signal of the released methane. The quantification of the archaeal and methanogenic community supports in large previous findings for other lake systems. The NGS gives some new insights into the community structure. Especially NGS data for the mcrA gene are currently still scarce in the literature. In addition they contrast the sediment of two lakes and show a well resolved depth profile of the top 20cm of the respective sediment.

Specific comments: Introduction: Line 73: belonging to the archaeal... Line 74: Methanogens from seven archaeal orders

Methods: Line 110: Were the five replicate cores taken at the same location or at different spots around the lake? Line 112: What was the diameter of the columnar sediment sampler? Line 114: 14.8°C not 14.8 °C Line 130ff: Conrad et al. 2010 used a time series to estimate the methane production potential as maximal slope of the methane concentration over the time for several consecutive points. (Compare Liu et al 2016). Using endpoint values will largely underestimate the methanogenic potential since most incubations will have a lag phase in the beginning without any methane production (compare Liu et al. 2016 in your references). Likewise the time span of 28 days may be insufficient to establish the full potential of such samples at the low incubation temperatures (16°C). Line 134: a total of six sediment... Line 134: Why did you initially mix the five cores (line118) and now redistribute into six replicates? Line 145 the quality of the DNA was checked... Line 151 mcrA and archaeal 16S rRNA genes, respectively (change order!) Line 155: The range of the standards is rather small. The results for 16S rRNA are not covered! Line 168: how was the quality filtering done? Results Line 199: The hydrogenotrophic methanogenic potential is not equal to the methane production under CH3F inhibition. CH3f partially inhibits the hydrogenotrophic methanogenesis as well; hence one has to use isotopic signals of the produced methane under both conditions together with dedicated fractionation factors to estimate the hydrogenotrophic contribution (Compare Conrad et al 201 as well as Liu et al 2016 in your references). Better use the term "inhibited samples" to

describe the methanogenic potential of these samples. Line 206: between these two... (delete: in) Discussion General: You tend to discuss several depth related changes by comparing your results to previous studies. You should carefully check (and quote) the respective sampling depth. You are doing a relatively well resolved profile, while many others use deeper cores.

Line 331: rate could differ drastically between the two... Line 350: How did you calculate the contribution of hydrogenotrophic methanogenesis? See comments to line 199? I would describe this more carefully! If you e.g. use the concentration values given in Conrad et al. 2010 for lake batata in Fig 1. (2.3 vs. 0.5 kPA) you would estimate a contribution of roughly 20%; using the isotope values you reach 30-50% (Table 5). Line 357: How does the produced methane correlate with the organic carbon in your study? Line 374 Mthanogen? Line 447 Methanombacteriales (last letter is currently not italic)

References: Bastviken et al. 2009 and Conrad et al 2014: incomplete:missing pages!

Tables and Figures Fig 1: The unit is nmol/g dry weight/day. However it is unclear how you have quantified the dry weight and it is very unlikely that your estimates using only endpoint values will give a meaningful estimate of the potential. I would rather show the amount of methane produced. Fig 2b: check the x axis! It somehow has different scaling than Fig 2a. Fig 3: Fig 4: The tree has only very few reference strains incorporated. Where are the Methanocellales? (Close to OUT 10 OUT 11 I would guess??). Where is Methanobacterialles Fig 5: I am not good in statistics but you seem to feed in much more data in the 16S tree than sequences in the mcrA tree. How does that influence the tree structure? Fig4: OUT 7 and OUT 1 which have the most sequences originate from E4 and E6 respectively? Supplementary Figures: Check Order: Fig S1 is first mentioned in line 341; while Fig S2 (Line 261) and S3 (Line 293) are mentioned much earlier.

Fig S2: error bars missing. Fig S2: you find a relative high relative contribution of methanogens in the top sediment sample; in contrast the activity there is apparently

low (Fig. 1). Likewise you find some sequences associated with Methanosarcinales; while in your NGS data (Fig S3) you do not find any Methanosarcinales? Fig S3: Give the clustername or related organisms in the figure legend (or legend) as well.

---

## Referee Comment (RC2) · A. Bagnoud (Referee) · 6 Sep 2016

Journal: BG Title: Vertical profiles of sediment methanogenic potential and communities in two plateau freshwater lakes Author(s): Y. Yang et al. MS No.: bg-2016-321 MS Type: Research article

1) General comments

This manuscript describes and compares methanogenesis in sediments from an eutrophic and a mesotrophic lake, with a focus on methanogens potential activity, abundance and diversity along vertical profiles. The results presented here are of interest. However, this manuscript could be significantly improved. Indeed, the current version

suffers from (i) a lack of general conclusion (i.e. the discussion and conclusion are superficial), (ii) a writing style that is sometimes confusing for the reader, (iii) poor-quality figures, and (iv) a lack of geochemical background. Concerning this last point, the geochemical background is available in the SI but totally absent from the main text and the discussion. This information could significantly help the discussion and the interpretation of the results.

2) Specific comments

a. Abstract

- l. 25-26: "changes of methanogens". Do you mean change in abundance or taxonomic change? Please be more specific.

- l. 35-36: For each lake, specify if it is mesotrophic or eutrophic.

- l. 33-43: This second part of the abstract looks like a list of results. It would be nice to have here some concluding remarks, i.e. put these results in the context of what we know about methanogenesis in these environments.

- l. 46-47: I am not sure that it makes sense to choose keywords that already appear in the title and/or in the abstract.

b. Introduction

- l. 77-78: This short description of methanogenesis pathways is over-simplified. Please expand or at least specify that those are the main pathways (and not the only ones).

- l. 81: What do you mean here by "theoretical ratio". Do you mean that we consider that, in most of natural environments, 2 molecules of $CH_4$ are produced by acetoclastic methanogen per molecule of $CH_4$ produced by hydrogenotrophic methanogens? Please be more specific.

- l.85-86: Does the 16S or mcrA sequencing give some insights about the methanogenesis pathway?

- l. 93: What is a humic lake? Could you please define it?

- l. 96: What do you mean by substrate here? Are you talking about the source of electron (hydrogen, acetate)?

c. Materials and methods

- l. 108-109: Here we are missing some background information about these 2 lakes. I propose the following: Indicate where these 2 lakes or located on a map, and describe them by listing some key features (that make these lakes eutrophic or mesotrophic, and/or that are relevant for this study). This information could be included in the figure containing the map.

- l. 113-114: Is this information relevant to the study?

- l. 120: The physicochemical analyses are simply absent from the manuscript. They should be included in the main text and more importantly in the discussion.

- l. 125-127: At this point, we don't want to know where to find the results, but how did you proceed to obtain these results.

- l. 137: What is the final CH3F concentration in the incubations?

- l. 130-141: How did you calculated the rates from these incubations?

- l. 144-161: What is the qPCR efficiency and how was inhibition tested?

- l. 166: What kind of triplicates are you talking about? Biological, technical?

- l. 170-171: "After subsampling to the lowest number of sequences". What do you mean here?

- l. 163-176: Did you check your sequences for chimeras?

- l. 174-176: Which method did you use for taxonomic annotation (Silva is only a

database)

- l. 180: Which method did you use for detecting chimeras?

- l. 186-187: The PCoA and environment clusters analysis are missing from the manuscript.

d. Results

- l. 195: I guess DW stands for dry weight. You should mention in the methods that the rates were calculated relative to the dry weight of sediments (and how you measured the dry weight of the sediments).

- l. 201: I think you mean HMP, not MPP.

- l. 204-207: It would be nice to see these results. A plot including MPP, AMP and HMP would help the reader to understand the results.

- l. 226: What is this normalization?

- l. 228: What is exactly the library coverage? I guess it's the proportion of 16S from the community that was sequenced. How was it calculated?

- l. 251: If you use this annotation, you should define it somewhere (in the Methods).

- l. 268-271: Which taxonomic order these genera belong to?

- l. 303-304: Which clustering method was used here?

e. Discussion

- l. 320-467: Here the structure chosen by the author for comparing their results with the ones from other previous studies looks a bit strange to me. First, they describe what was observed in other studies, and then, in a second sentence, they summarize what they found. This is confusing for the reader. I would do it the other way around, i.e. "we observed this, which is consistent (or not) with previous observations". - l. 320: In which kind of environment are you referring to?

- l. 334-335: What do you mean by geological constitute, geographic regions and water type. Please be more specific and/or develop a little bit.

- l. 339-341: What is the correlation between TOC (and other environmental parameters) and MPP, HMP and AMP?

- l. 343-346: Did you find similar OTUs between these 2 lakes sediments?

- l. 357: Please develop.

- l. 359-361: I don't understand what the authors mean here.

- l. 365-366: This is not clear. Should we conclude that shallow and eutrophic lakes are not stratified?

- l. 370-371: Please develop here a bit more.

- l. 373-374: How is it possible to assess the abundance of methanotrophs with qPCR based on archaeal 16S primers? This makes no sense to me.

- l. 377: Consistent with which results?

- l. 379-380: What is the range of archaeal abundance described in the literature?

- l. 383: Different archaeal primers combinations were used for 16S quantification and sequencing. That could be an explanation for the different ratios observed.

- l. 383-384: Do you mean that organisms can have multiple copies of mcrA? Or of 16S rRNA gene? Not really clear. Please develop and back-up with genomic data from literature.

- l. 394-395: "but the change pattern. ... was not clear". I don't understand it. Can you clarify?

- l. 397-398: Can you point out the results you are discussing here?

- l. 418: Could you briefly introduce Taihu Lake?

[Figure]

- l. 420: "remarkably" is too strong here. I don't find this result very surprising...

- l. 420: Please point out a figure to illustrate these findings.

- l. 412: Where do you see that depth is a key factor for archaeal community structure?

- l. 425-426: It seems that you are describing quantitative changes. But are they qualitative changes as well (at the genus level)?

- l. 437-440: I would like to see this data.

- l. 442: What do you mean by "the seventh"? The 7th to be discovered or the 7th most important? Not clear.

- l. 454-461: The congruency of phylogenetic trees does not depend on the relative proportion of the different OTUs. In fact, this has not impact. Do the 2 different trees have the same shape? Are the clusters the same between these 2 trees?

f. Figures

- Fig. 1 and 2: Flip over the axis to have the depth as vertical and descending axis. Precise which lake is eutrophic and which one is mesotrophic.

- Fig. 1: Merge the 2 plots so it's easier to compare the results. Indicates MPP, AMP and HMP, and the AMP/HMP ratio.

- Fig. 2: Use log scale for gene abundance.

- Fig. 3: You should make this figure more informative. I suggest the following: you could start from the genus level. You indicate only genera that represent at least few % of the archaeal community (define a threshold) and merge the ones below this threshold in a category called "other <family name>". Then you do the same with the next taxonomic level (family), merging the small ones into the higher level (i.e. class), and so one. It's a little bit more of work to produce this figure, but at the end you have a plot that indicates the dominating groups, independently of the taxonomic level. An

alternative is the Krona charts (https://sourceforge.net/p/krona/home/krona/).

- Fig. 4: Highligh the E and D samples with a color code.

- Fig. 5: The horizontal distances are not defined. Please use a color-code as well.

3) Technical corrections

a. Abstract

- l. 25: "layer depth" is confusing. Please change it for "depth" throughout the whole document.

b. Results

- l. 191-193: These 2 first sentences are not clear. You should re-write them as follow: "In this study, MPP varied remarkably with both lake and sediment depth except for the uppermost sediments layers which are remarkably similar between the two lakes (Figure 1)."

- l. 213: Remove "However".

- l. 214: Rephrase "and showed an increase with depth..."

- l. 220: Which depth these samples (D6 and E5) correspond to?

c. Discussion

- l. 379: Please rephrase: "...the abundance of archaea..."

- l. 381: Specify here that this first ratio was calculated with qPCR results.

---

## Author Comment (AC1) · 12 Oct 2016

The manuscript "Vertical profile of sediment methanogenic potential and communities in two plateau freshwater lakes" is comprehensive written. The aim of the study is clearly stated and well supported with data. The authors describe the methanogenic potential (MPP) of sediment incubations, quantify the archaeal and methanogenic community and analyses the community structure using NGS. They can show that the two lakes exhibit different patterns for almost all analyzed parameters and show some changes along a depth profile of 20cm. The MPP measurements would benefit from a better time resolution and the additional measurement of the isotopic signal of the released methane. The quantification of the archaeal and methanogenic community

supports in large previous findings for other lake systems. The NGS gives some new insights into the community structure. Especially NGS data for the mcrA gene are currently still scarce in the literature. In addition they contrast the sediment of two lakes and show a well resolved depth profile of the top 20cm of the respective sediment.

ResponseïijŽThe authors appreciate the reviewer's positive comments. Revisions were made in response to the following points.

Specific comments: Introduction: Line 73: belonging to the archaeal. . . Response: The authors appreciate the reviewer's suggestions. As suggested, the revision has been made.

Line 74: Methanogens from seven archaeal orders Response: As suggested, the revision has been made.

Methods: Line 110: Were the five replicate cores taken at the same location or at different spots around the lake? Response: They were taken at the same location. Preliminary experiement has been done to test the methane production potential in surface (top 5 cm) sediment of several sites around the lake, and the site with a median MPP was chosen for the current study.

Line 112: What was the diameter of the columnar sediment sampler? Response: 10 cm.

Line 114: 14.8o C not 14.8C Response: As suggested, the revision has been made.

Line 130ff: Conrad et al. 2010 used a time series to estimate the methane production potential as maximal slope of the methane concentration over the time for several consecutive points. (Compare Liu et al 2016). Using endpoint values will largely underestimate the methanogenic potential since most incubations will have a lag phase in the beginning without any methane production (compare Liu et al. 2016 in your references). Likewise the time span of 28 days may be insufficient to establish the full potential of such samples at the low incubation temperatures (16oC).

Response: The authors appreciate the reviewer's interesting points. In the preliminary experiment, methane production was measured at an interval of one week, and result showed that the methane concentration could be detected at the end of the first week (although quite low), and remained nearly constant after 5 weeks. Hence, in the current study, methane concentration was measured at the end of the 4th week to get an approximate methane production potential. To avoid the inaccurate expression, we use total methane production instead of methane production potential in the new version of manuscript.

Line 134: a total of six sediment... Response: As suggested, the revision has been made.

Line 134: Why did you initially mix the five cores (line 118) and now redistribute into six replicates? Response: One sediment core was not enough for all the analysis. To attain a sufficient amount of sediment sample, we had to take several cores. Considering there might be difference between these cores, they were mixed and redistributed for physicochemical, activity and molecular analysis.

Line145 the quality of the DNA was checked... Response: As suggested, the revision has been made.

Line 151 mcrA and archaeal 16S rRNAgenes, respectively (change order!) Response: As suggested, the revision has been made.

Line 155: The range of the standards is rather small. The results for 16S rRNA are not covered! Response: We feel sorry that we made a mistake of the unit in the manuscript. It has been corrected as copies/$\mu$L.

Line 168: how was the quality filtering done? Response: By using the Sliding Window quality filtering (Trimmomatic). The window was set as 50 bp, and the threshold was set as 20.

Results Line 199: The hydrogenotrophic methanogenic potential is not equal to

the methane production under CH3F inhibition. CH3F partially inhibits the hydrogenotrophic methanogenesis as well; hence one has to use isotopic signals of the produced methane under both conditions together with dedicated fractionation factors to estimate the hydrogenotrophic contribution (Compare Conrad et al 201 as well as Liu et al 2016 in your references). Better use the term "inhibited samples" to describe the methanogenic potential of these samples. Response: The authors appreciate the reviewer's suggestions. As suggested, the revision has been made.

Line 206: between these two...(delete: in) Response: The authors appreciate the reviewer's suggestions. As suggested, the revision has been made.

Discussion General: You tend to discuss several depth related changes by comparing your results to previous studies. You should carefully check (and quote) the respective sampling depth. You are doing a relatively well resolved profile, while many others use deeper cores. Response: The authors appreciate the reviewer's suggestions. Most studies we chose for comparison have a comparable sampling depth with ours, especially those for activity and abundance are all limited to less than 50 cm. To put it more clearly, we have added the respective sampling depth as well as interval to the manuscript as suggested.

Line 331: rate could differ drastically between the two... Response: The authors appreciate the reviewer's suggestions. As suggested, the revision has been made.

Line 350: How did you calculate the contribution of hydrogenotrophic methanogenesis? See comments to line 199? I would describe this more carefully! If you e.g. use the concentration values given in Conrad et al. 2010 for lake batata in Fig 1. (2.3 vs. 0.5 kPA) you would estimate a contribution of roughly 20%; using the isotope values you reach 30-50% (Table 5).

Response: The authors appreciate the reviewer's suggestions. Since isotope data was lacking, the contribution might not be accurately estimated. We had replaced AMP with IMP (inhibited methane production). Conrad et al (2010) reported the CH3F inhibited

about 90% of hydrogenotrophic methanogenesis, so we might estimate the contribution roughly.

Line357: How does the produced methane correlate with the organic carbon in your study? Response: The TMP showed a significant correlation with sediment TOC (Spearman rank correlation, $p<0.05$). However, methane production might be related to several parameters (TOC, ORP and so on), and most of these parameters were also related to depth. So a simple correlation might give misleading conclusion. Unluckily, data here was not enough for partial correlation. So correlation analysis was not shown here, and the correlation was not discussed in the manuscript.

Line 374 Mthanogen? Response: It has been corrected as methanogen.

Line 447 Methanombacteriales (last letter is currently not italic) Response: As suggested, the revision has been made.

References: Bastviken et al. 2009 and Conrad et al 2014: incomplete:missing pages! Response: The authors appreciate the reviewer's suggestions. As suggested, the revision has been made.

Tables and Figures Fig 1: The unit is nmol/g dry weight/day. However it is unclear how you have quantified the dry weight and it is very unlikely that your estimates using only endpoint values will give a meaningful estimate of the potential. I would rather show the amount of methane produced. Response: The authors appreciate the reviewer's suggestions. As suggested, the revision has been made.

Fig 2b: check the x axis! It somehow has different scaling than Fig 2a. Response: The authors appreciate the reviewer's suggestions. As suggested, the revision has been made.

Fig 4: The tree has only very few reference strains incorporated. Where are the Methanocellales? (Close to OUT 10 OUT 11 I would guess??). Where is Methanobacterialles Response: The authors appreciate the reviewer's suggestions. Several

reference strains had been added to the tree. Methanocellales strain NC009464 (Methanocella arvoryzae) was related to OUT 23-28. And Methanobacterialles strains Methanobacterium beijingense and Methanobacterium congolense were also shown in the tree.

Fig 5: I am not good in statistics but you seem to feed in much more data in the 16S tree than sequences in the mcrA tree. How does that influence the tree structure? Response: This doesn't significantly change the tree structure. The mcrA libraries have a high coverage, indicating that the diversity has been well captured.

Fig4: OUT 7 and OUT 1 which have the most sequences originate from E4 and E6 respectively? Response: OTU 7 has the more sequences originate from both E4 and E6. The representative sequence was attained using Mothur command Get.oturep, and was not related to the relative abundance of different samples.

Supplementary Figures: Check Order: Fig S1 is first mentioned in line 341; while Fig S2 (Line 261) and S3 (Line 293) are mentioned much earlier. Response: As suggested, the revision has been made.

Fig S2: error bars missing. Response: Replicated samples were mixed and then analyzed, so here we only gave an average result.

Fig S2: you find a relative high relative contribution of methanogens in the top sediment sample; in contrast the activity there is apparently low (Fig. 1). Likewise you find some sequences associated with Methanosarcinales; while in your NGS data (Fig S3) you do not find any Methanosarcinales? Response: A high relative abundance might not always consist with activity. In our result, a high relative contribution of methanogens was found in the top layer of Erhai Lake, and the activity was high, too. In contrast, methanogen showed high abundance but low activity (still higher than the top layer of Erhai Lake) in the top layer of Dianchi Lake. In the NGS data, Methanosarcinales were found in all samples.

Fig S3: Give the cluster name or related organisms in the figure legend (or legend) as well. Response: The authors appreciate the reviewer's suggestions. As suggested, the revision has been made.

---

## Author Comment (AC2) · 13 Oct 2016

Response to Referee #2's comments 1) General comments This manuscript describes and compares methanogenesis in sediments from a eutrophic and a mesotrophic lake, with a focus on methanogens potential activity, abundance and diversity along vertical profiles. The results presented here are of interest. However, this manuscript could be significantly improved. Indeed, the current version suffers from (i) a lack of general conclusion (i.e. the discussion and conclusion are superficial), (ii) a writing style that is sometimes confusing for the reader, (iii) poor-quality figures, and (iv) a lack of geochemical background. Concerning this last point, the geochemical background is available in the SI but totally absent from the main text and the discussion. This information could significantly help the discussion and the interpretation of the results. Response: The authors appreciate the reviewer's valuable comments. In the revised manuscript, (i) General conclusion has been added to the manuscript, (ii) writing has been improved, (iii) some figures have been redrawn as suggested, (iv) geochemical background of the two lakes has been added to the main text.

2) Specific comments a. Abstract - l. 25-26: "changes of methanogens". Do you mean change in abundance or taxonomic change? Please be more specific. Response: The authors appreciate the reviewer's suggestions. The revision has been made as follows: "knowledge on the layer depth-related changes of methanogen community structure and activities in freshwater lake sediment..."

- l. 35-36: For each lake, specify if it is mesotrophic or eutrophic. Response: The revision has been made as follows: "The layer depth-related change pattern of the methanogenesis potential in eutrophic Dianchi Lake was found to be different from that in mesotrophic Erhai Lake."

- l.33-43: This second part of the abstract looks like a list of results. It would be nice to have here some concluding remarks, i.e. put these results in the context of what we know about methanogenesis in these environments. Response: The authors appreciate the reviewer's valuable suggestions. The revision has been made as suggested.

- l.46-47: I am not sure that it makes sense to choose keywords that already appear in the title and/or in the abstract. Response: The authors appreciate the reviewer's suggestions. 'vertical profiles' has been removed from the keywords, and 'trophic status' has been added.

b. Introduction - l. 77-78: This short description of methanogenesis pathways is oversimplified. Please expand or at least specify that those are the main pathways (and not the only ones). Response: The authors appreciate the reviewer's suggestions. The revision has been made as follows: In freshwater lake, organic matter is fermented to acetate, $CO_2$ and $H_2$, which are further converted to $CH_4$ by methanogens. There

are two major pathways: hydrogenotrophic pathway (using H2/CO2) and acetoclastic pathway (using acetate, i.e. the methyl group) (Conrad et al., 2010). And the relative contribution of the two pathways varies in different lakes (Conrad, 1999).

- l. 81: What do you mean here by "theoretical ratio". Do you mean that we consider that, in most of natural environments, 2 molecules of CH4 are produced by acetoclastic methanogen per molecule of CH4 produced by hydrogenotrophic methanogens? Please be more specific. Response: The authors appreciate the reviewer's suggestions. The revision has been made as follows: Despite the theoretical ratio of 2:1 (acetoclastic pathway: hydrogenotrophic pathway) when carbohydrates or other similar form of organic matter was degraded. . .

- l.85-86: Does the 16S or mcrA sequencing give some insights about the methanogenesis pathway? Response: Yes. Certain genus of methanogens might be related to certain pathway. For instance, the order Methanomicrobiales dominated in most samples, and it was hydrogenotrophic.

- l. 93: What is a humic lake? Could you please define it? Response: In the reference, it generally refers to a lake which contains darkly stained acid water.

- l. 96: What do you mean by substrate here? Are you talking about the source of electron (hydrogen, acetate)? Response: Organic matter (the same as mentioned in the references).

c. Materials and methods - l. 108-109: Here we are missing some background information about these 2 lakes. I propose the following: Indicate where these 2 lakes or located on a map, and describe them by listing some key features (that make these lakes eutrophic or mesotrophic, and/or that are relevant for this study). This information could be included in the figure containing the map. Response: The authors appreciate the reviewer's valuable suggestion. A table has been added to provide essential background.

- l. 113-114: Is this information relevant to the study? Response: Yes. The incubation temperature was set close to the in-situ temperature.

- l. 120: The physicochemical analyses are simply absent from the manuscript. They should be included in the main text and more importantly in the discussion. Response: The authors appreciate the reviewer's suggestions. The physicochemical characteristics of the two lakes have been well studied in a series of articles (e.g. a 26-article thematic issue on Environmental Earth Sciences, volume 74, issue 5). So here we just provide a general background of Dianchi Lake and Erhai Lake. We have added some key points as suggested. The methanogen community structure and activity are always thought to be related physicochemical parameters of course. However, in vertical profiles, both physicochemical parameters (e.g. DOC, TN, ORP) and microbial community change with depth. The co-variation makes correlation analysis doubtable. That is to say, though we observed significant positive correlation between TMP and sediment $NH_4^+$-N, TP and TOC in our study, we could not draw a conclusion that TMP was impacted by these parameters. There still remains possibility that depth has similar impact on TMP and these parameters so they just show a similar change pattern.

- l. 125-127: At this point, we don't want to know where to find the results, but how did you proceed to obtain these results. Response: The authors appreciate the reviewer's suggestions. The revision has been made as suggested.

- l. 137: What is the final $CH_3F$ concentration in the incubations? Response: 2%.

- l. 130-141: How did you calculated the rates from these incubations? Response: In the new version of manuscript, the rate has been replaced by total methane production.

- l. 144-161: What is the qPCR efficiency and how was inhibition tested? Response: qPCR efficiency was 96.52% for Archaeal 16S rRNA gene, and 92.93% for mcrA gene. Inhibition was tested using Cq dilution method.

- l. 166: What kind of triplicates are you talking about? Biological, technical? Response: Biological.

- l. 170-171: "After subsampling to the lowest number of sequences". What do you mean here? Response: Since different numbers of sequences were obtained from NGS, calculating the diversity index (maybe as well as some other parameters) using all the raw data might result in bias (i.e. higher number of sequences might results in higher diversity). So sequences in each sample were random sampled to a same number (to make each sample contain same number of sequences).

- l. 163-176: Did you check your sequences for chimeras? Response: Sorry that we made a mistake at line 180. We checked our archaeal 16S rRNA sequences (instead of mcrA sequences) for chimeras during the cluster process.

- l. 174-176: Which method did you use for taxonomic annotation (Silva is only a database) Response: Naive Bayesian classifier (http://sourceforge.net/projects/rdp-classifier/).

- l. 180: Which method did you use for detecting chimeras? Response: Sorry that we made a mistake at line 180. We checked our archaeal 16S rRNA sequences (instead of mcrA sequences) for chimeras during the cluster process.

- l. 186-187: The PCoA and environment clusters analysis are missing from the manuscript. Response: The revision has been made as follows: . . .and cluster analysis were conducted based on Weighted Unifrac distance. . . The result was shown in figure 5.

d. Results - l. 195: I guess DW stands for dry weight. You should mention in the methods that the rates were calculated relative to the dry weight of sediments (and how you measured the dry weight of the sediments). Response: The authors appreciate the reviewer's suggestions. The revision has been made as follows: (DW stands for dry weight, which was determined gravimetrically). . .

- l. 201: I think you mean HMP, not MPP. Response: The authors appreciate the

reviewer's suggestions. The revision has been made.

- l. 204-207: It would be nice to see these results. A plot including MPP, AMP and HMP would help the reader to understand the results. Response: The authors appreciate the reviewer's suggestions. The revision has been made.

- l. 226: What is this normalization? Response: In the new version, it has been replaced by subsampling (as is described in line 170-171).

- l. 228: What is exactly the library coverage? I guess it's the proportion of 16S from the community that was sequenced. How was it calculated? Response: Library coverage is an estimation of the proportion of genes from the community that was sequenced. And it was calculated as C=1-n/N, where n is the number of OTUs without a replicate, and N is the total number of sequences.

- l. 251: If you use this annotation, you should define it somewhere (in the Methods). Response: The authors appreciate the reviewer's suggestions. The revision has been made as: The five replicate sediment cores were sliced into the layers (sample D1 or E1:0–5 cm, sample D2 or E2:5–8 cm, sample D3 or E3:8–11 cm, sample D4 or E4:11–14 cm, and sample D5 or E5:14–17 cm, sample D6 or E6:17–20 cm). Samples D1–D6 and E1–E6 were from Dianchi Lake and Erhai Lake, respectively.

- l. 268-271: Which taxonomic order these genera belong to? Response: The revision has been made as follows: At genus level, Methanobacterium (within Methanobacteriales) had the greatest proportion, followed by Methanosaeta (within Methanosarcinales) and Methanoregula (within Methanomicrobiales)

- l. 303-304: Which clustering method was used here? Response: UPMGA

e. Discussion - l. 320-467: Here the structure chosen by the author for comparing their results with the ones from other previous studies looks a bit strange to me. First, they describe what was observed in other studies, and then, in a second sentence, they summarize what they found. This is confusing for the reader. I would do it the

other way around, i.e. "we observed this, which is consistent (or not) with previous observations". Response: The authors appreciate the reviewer's suggestions. The revision has been made as suggested.

- l. 320: In which kind of environment are you referring to? Response: The revision has been made as follows: The methanogenesis potential in freshwater lakes varied in a wide range

- l. 334-335: What do you mean by geological constitute, geographic regions and water type. Please be more specific and/or develop a little bit. Response: The revision has been made as follows: . . .including geological constitute (e.g. calcareous or not), geographical regions (Rinta et al., 2015) as well as water type (e.g. black water, clear wate) (Conrad et al., 2014). . .

- l. 339-341: What is the correlation between TOC (and other environmental parameters) and MPP, HMP and AMP? Response: Methane production might be related to several parameters (TOC, ORP and so on), and most of these parameters were also related to depth. So simple correlation would give misleading conclusion. Unluckily, data here was not enough for partial correlation. So correlation analysis was lacked here.

- l. 343-346: Did you find similar OTUs between these 2 lakes sediments? Response: Yes. The major OTUs (e.g. OTU1, OTU7) showed notable similarity.

- l. 357: Please develop. Response: The authors appreciate the reviewer's suggestions. The revision has been made as suggested.

- l. 359-361: I don't understand what the authors mean here. Response: The authors appreciate the reviewer's suggestions. The revision has been made.

- l. 365-366: This is not clear. Should we conclude that shallow and eutrophic lakes are not stratified? Response: Yes, shallow lakes are not stratified.

- l. 370-371: Please develop here a bit more. Response: The authors appreciate the

reviewer's suggestions. The revision has been made as suggested.

- l. 373-374: How is it possible to assess the abundance of methanotrophs with qPCR based on archaeal 16S primers? This makes no sense to me. Response: Some researchers have used 16S rRNA primers targeting certain orders of methanogens. The revision has been made as: The abundance of methanogens could be assessed using either order-specific archaeal 16S rRNA gene primers or mcrA gene primers.

- l. 377: Consistent with which results? Response: the results in the mentioned references.

- l. 379-380: What is the range of archaeal abundance described in the literature? Response: Response: The authors appreciate the reviewer's suggestions. The revision has been made as: In the current study, the abundance of archaeal 16S rRNA gene was comparable to that reported in the literatures (about $1\times107–2\times109$ copies/gDW in the top 20 cm) (Borrel et al., 2012; Zhu et al., 2012).

- l. 383: Different archaeal primers combinations were used for 16S quantification and sequencing. That could be an explanation for the different ratios observed. Response: The authors appreciate the reviewer's suggestions. The revision has been made as: The result might suggest either the bias of amplification of mcrA gene or the difference between the two archaeal 16S rRNA primers used in the current study.

- l. 383-384: Do you mean that organisms can have multiple copies of mcrA? Or of 16S rRNA gene? Not really clear. Please develop and back-up with genomic data from literature. Response: The authors appreciate the reviewer's suggestions. The revision has been made as: The result might suggest either the bias of amplification of mcrA gene or the difference between the two archaeal 16S rRNA primers used in the current study.

- l. 394-395: "but the change pattern: : :. was not clear". I don't understand it. Can you clarify? Response: The authors appreciate the reviewer's suggestions. The revision

has been made as follows: but the mcrA gene diversity didn't show a regular change pattern

- l. 397-398: Can you point out the results you are discussing here? Response: The authors appreciate the reviewer's suggestions. The revision has been made as follows: In this current study, the sediment samples from Erhai Lake had slightly lower mcrA gene diversity than those from Dianchi Lake (Table 1).

- l. 418: Could you briefly introduce Taihu Lake? Response: As described in the reference, Lake Taihu is the third largest freshwater lake in China. It is s shallow eutrophic lake with an average depth of about 2 m.

- l. 420: "remarkably" is too strong here. I don't find this result very surprising. . . Response: The authors appreciate the reviewer's suggestions.The revision has been made as follows: . . .archaeal community structure differed evidently in Dianchi Lake and Erhai Lake. . ..

- l. 420: Please point out a figure to illustrate these findings. Response: The authors appreciate the reviewer's suggestions.The revision has been made as follows: In this study, the result of UniFrac-based cluster analysis (Figure 5a). . .

- l. 412: Where do you see that depth is a key factor for archaeal community structure? Response: (If the referee means line 421 here.) Figure 5a showed that samples from upper layers (D1-D3, E1) were separated from others.

- l. 425-426: It seems that you are describing quantitative changes. But are they qualitative changes as well (at the genus level)? Response: We meant to describe qualitative changes. They qualitatively changed as well.

- l. 437-440: I would like to see this data. Response: Data was shown in Figure S3.

- l. 442: What do you mean by "the seventh"? The 7th to be discovered or the 7th most important? Not clear. Response: The revision has been made as follows: "and was the seventh order of methanogens to be found".

- l. 454-461: The congruency of phylogenetic trees does not depend on the relative proportion of the different OTUs. In fact, this has not impact. Do the 2 different trees have the same shape? Are the clusters the same between these 2 trees? Response: As reported by Luton (2002), the 2 different tree should have the sample shape and the same clusters.

f. Figures - Fig. 1 and 2: Flip over the axis to have the depth as vertical and descending axis. Precise which lake is eutrophic and which one is mesotrophic. Response: The authors appreciate the reviewer's suggestions. As suggested, the revision has been made.

- Fig. 1: Merge the 2 plots so it's easier to compare the results. Indicates MPP, AMP and HMP, and the AMP/HMP ratio. Response: The authors appreciate the reviewer's suggestions. As suggested, the revision has been made.

- Fig. 2: Use log scale for gene abundance. Response: The authors appreciate the reviewer's suggestions. As suggested, the revision has been made.

- Fig. 3: You should make this figure more informative. I suggest the following: you could start from the genus level. You indicate only genera that represent at least few % of the archaeal community (define a threshold) and merge the ones below this threshold in a category called "other <family name>". Then you do the same with the next taxonomic level (family), merging the small ones into the higher level (i.e. class), and so one. It's a little bit more of work to produce this figure, but at the end you have a plot that indicates the dominating groups, independently of the taxonomic level. An alternative is the Krona charts (https://sourceforge.net/p/krona/home/krona/). Response: The authors appreciate the reviewer's suggestions. It would be nice if we could use Krona charts here. However, it seems there is not enough space for 12 pie charts. We have considered using heatmap as well, but most genus are of low abundance, and heatmap couldn't show the structure well. Considering that we mainly focus on an overall composition and methanogens in the current study, other details are omitted in

[Figure]

the manuscript.

- Fig. 4: Highligh the E and D samples with a color code. Response: The authors appreciate the reviewer's suggestions. As suggested, the revision has been made.

- Fig. 5: The horizontal distances are not defined. Please use a color-code as well. Response: The authors appreciate the reviewer's suggestions. As suggested, the revision has been made.

3) Technical corrections a. Abstract - l. 25: "layer depth" is confusing. Please change it for "depth" throughout the whole document. Response: The authors appreciate the reviewer's suggestions. As suggested, the revision has been made.

b. Results - l. 191-193: These 2 first sentences are not clear. You should re-write them as follow: In this study, MPP varied remarkably with both lake and sediment depth except for the uppermost sediments layers which are remarkably similar between the two lakes Figure 1)." Response: As suggested, the revision has been made.

- l. 213: Remove "However". Response: As suggested, the revision has been made.

- l. 214: Rephrase "and showed an increase with depth: : :" Response: As suggested, the revision has been made.

- l. 220: Which depth these samples (D6 and E5) correspond to? Response: The revision has been made as follows: . . .while the lowest one occurred in Dianchi Lake sample D6 (17-20cm, $2.5\pm0.3\times10^4$ copies/g dry weight) or Erhai Lake sample E5 (14-17 cm, $3.7\pm0.1\times10^4$ copies/g dry weight).

c. Discussion - l. 379: Please rephrase: ": : :the abundance of archaea: : :" Response: The revision has been made as follows: In the current study, the abundance of archaeal 16S rRNA gene

- l. 381: Specify here that this first ratio was calculated with qPCR results. Response: The revision has been made as follows: . . .for each sample, the mcrA/16S ratio was

less than 3% according to the results of qPCR

---

## Referee Report (RR1)

This manuscript describes the methanotrophic activity along the profile of sediments in an eutrophic and a mesotrophic lake. Hydrogenotrophic and acetoclastic methanogenesis was measured in lab incubation. Molecular analyses were also carried out, in order to quantify and describe the diversity of Archaea and methanogens. It is the second time I review this paper, and this version is significantly better than the previous one. But I still find the discussion part a bit weak. I would like more connections between the incubation and the molecular analyses. For instance, are they any correlation between the activity measurements and the concentration in the sediment of methanogens, or one clade of methanogens. This is missing and this piece of information could be valuable.

Below are some specific comments:

-25: "ecosystems" instead of "ecosystem"

-25: "layer depth-related" is unclear to me. Why not replace it by "depth-related"? Please replace "layer-depth", by "depth" or by "profile" throughout the whole document.

-29-30: Rephrase: "in two freshwater lakes at different trophic status on the Yunnan Plateau (China), Dianchi Lake (eutrophic) and Erhai Lake (mesotrophic)."

-34-36: Remove: "The layer depth-related changes of methanogenesis potential, and the abundance and community structure of methanogens were observed in either Dianchi Lake or Erhai Lake."

-36-37: Rephrase: "…mcrA and archaeal 16S rRNA genes displayed a similar abundance change…"

-42: Only genus and species names should be italicized (http://blog.vancouvereditor.com/2011/03/science-writing-and-editing-scientific.html)

-44: Which trophic status? Please introduce it in the abstract, for instance in the sentence of l. 29-30.

-45: You did not describe the vertical change in methanogenic activity in this abstract. Please add these results in the abstract.

-48-49: It is not necessary to use keywords that are already present in the abstract (such as mcrA).

-84: remove a space here: "acetoclastic pathway:hydrogenotrophic pathway"

-85: Typo: "similar forms of organic matter"

-89: Please rephrase: "To identify the methanogens and methanogenic pathways, both archaeal 16S rRNA…"

-97: Remove "Many". You mention 1 or 2 studies per type of lake, which is not a lot.

-100: Please clarify in the text what do you mean by "substrate".

-102-103: "Methanogensis … might be different". What do you mean here? Different in activity, community structure...? Please clarify.

-105: Correct this way: "… our attention: (1) How…"

-122: Write "in situ" in italics

-127-129: I know that it is quite obvious that D stands for Dianchi and E for Erhai, but you should state it explicitly somewhere.

-146: We don't really care about the volume of your CH3F solution you added to your incubation flasks. We want to know what is the final concentration. Please add this piece of information (2%).

-154: Replace "quality" by "fragment size"

-163: "literature" instead of "literatures".

-166-167: Please specify in the text the coefficient of efficiency (E) and if inhibition tests were conducted.

-168: For the ANOVA, were the data normally distributed between the 2 groups? Usually, we log-transform qPCR data for this kind of statistical test. Did you do it? More details are needed here.

-175: You told me that this triplicate was a biological one. How it is possible, knowing to fact that you pooled together the 5 replicates you sampled, as stated in lines 131-132?

-178-179: You have to deposit the 16S sequences to GenBank as well.

-179-180: "After subsampling to the lowest number of sequences". This is not clear. Please rephrase.

-184: "affiliation" instead of "identities"

-184-185: More details about the taxonomic annotations are needed. How did you compare your sequences to the database, which similarity threshold did you choose?

-187-189: You have to deposit the raw reads to SRA as well.

-192-193: More details about the phylogenetic analysis are needed: which model? Bootstrap? Tree method?

-207: Which statistical test did you use here? Please add it to the methods section.

-212-213: Please rephrase "Dianchi Lake had the IMP of 47.0–182.8 nmol/gDW" to "IMP in Dianchi Lake ranges from… to…"

-216-217: In which lake is the acetoclastic methanogensis more important? Did you perform any statistical test to assess this (such as the one used in l. 207)? Please clarify in the text.

-224-225: Please describe in the text the stats behind this p-value.

-238: Please quickly define "library coverage" for the reader.

-271-282: I strongly recommend to switch figure 3 with figure S1. Indeed, the content of figure S1 is directly related to the focus of the article, while figure 3 is not. Indeed, figure S1 is more relevant and meaningful to the research topic than figure 3. Same thing with figure S2, I think it deserves to be in the main text.

-316: Please add here the clustering method (UPGMA)

-333: "literature" instead of "literatures"

-357-359: Use the present tense in this sentence.

-369-370: I don't understand the link between the acetoclastic methanogenesis and the availability of organic matter. Both methanogenic pathways use as substrate fermented organic matter. Why the availability of organics would have a greater influence on the acetoclastic pathway? Could you please clarify it in the text?

-370-373: Did you try to correlate acetoclastic methanogenesis with organic matter content?

-390-391: Please add a link to a figure.

-419-432: The high proportion of Methanomicrobiales (hydrogenotrophic) and the low proportion of Methanosarcinales (acetoclastic) is in agreement with your incubations, which highlighted a dominance of the hydrogenotrophic methanogenesis, over the acetoclastic one. You should write it somewhere in the main text.

-425: Please add the description of Taihu Lake in the text (i.e. the third largest freshwater lake in China. It is s shallow eutrophic lake with an average depth of about 2 m.)

-437-442: Do you have some geochemical data (maybe from other studies) that could explain these abrupt changes in the sediment profiles? If so, please add them to the discussion.

-448-450: Please refer to figure S1 to show the data.

-450-453: Please refer to figure S2 to show the data.

-459-461: Please rephrase this way: "Methanoplasmatales-like methanogens are related to Thermoplasmatales archaeons. They are usually present in termite guts and high-salinity environments."

-464-465: Please refer to figure S1.

-Figure 1: Please put the horizontal axis at the top.

-Figure 2: Specify in the caption and/or in the label axis the log scale of the gene abundance.

-Figure 2: Please inverse the vertical axis for a better readability. Put the horizontal axis at the top.

-Figure 4 caption: I think the bar represents 10% divergence.

-Figures S1 and S2 captions: "Samples D1–D6 and E1–E6 were from Dianchi Lake and Erhai Lake, respectively." was written 2 times.

-Figure S3: I would set the depth as a descending vertical axis (just like in figure 1 and 2).

-Missing things in this article:

  - OTU tables (16S and mcrA-based) should be available to the reader (should be published online).

  - Are there any correlation between data from figure 1 and figure S1/S2. Can you explain the activity profiles with your molecular data?

- Why is there only a tree with the mcrA data. Did you try to build a tree with 16S data? Are both trees congruent?